# A Practical and Stealthy Adversarial Attack for Cyber-Physical Applications

## Yifu Wu, Jin Wei-Kocsis

Department of Computer and Information Technology
401 N Grant Street
West Lafayette, Indiana 47907
{wu1584, kocsis0}@purdue.edu

## Abstract

Adversarial perturbations on misleading a well-trained machine learning (ML) model have been studied in computer vision (CV) and other related application areas. However, there is very limited focus on studying the impact of adversarial perturbations on ML models used in data-driven cyber-physical systems (CPSs) that normally have complex physical and mechanical constraints. Because of the complex physical and mechanical constraints, called domain-knowledge constraints in our paper, established gradient-based adversarial attack methods are not always practical in CPS applications. In this paper, we propose an innovative CPS-specific adversarial attack method that is able to practically compromise the ML-based decision makings of CPSs while maintaining stealthy by meeting the complex domain-knowledge constraints. In the section of performance evaluations, different scenarios are considered to illustrate the effectiveness of the proposed adversarial attack method in achieving a high success rate as well as sufficient stealthiness in CPS applications.

## 1 Introduction

In recent years, increasing evidence shows that carefully crafted adversarial perturbation is able to introduce bounded subtle adversarial perturbation that can mislead learning models to achieve incorrect decision making (Szegedy et al. 2013). Extensive research has been developed to study the impact of adversarial attack in different data-driven applications (Krizhevsky, Sutskever, and Hinton 2012; Lin et al. 2017; Vaswani et al. 2017; Goodfellow, Shlens, and Szegedy 2014; Alzantot et al. 2018; Alzantot, Balaji, and Srivastava 2018; Carlini and Wagner 2017). From the existing studies, it is clear that the vulnerability raised by adversarial attacks makes ML models not always trustworthy when being deployed in real-world applications such as self-driving car, face recognition, and Q&A systems. Therefore, it is crucial to sufficiently mitigate the adversarial perturbations. To realize successful mitigation strategy, it can be beneficial to first exploit adversarial mindset and formulate threat models of practical adversarial perturbations for a given application field.

To achieve this goal, many techniques have established to generate adversarial perturbations successfully misleading ML models on CV- and NLP- related application fields (Szegedy et al. 2013; Goodfellow, Shlens, and Szegedy 2014; Carlini and Wagner 2017; Devlin et al. 2018; Zhang et al. 2020). However, there is very limited work focusing on generating practical adversarial attacks in CPS-related application domain. A CPS, such as power system and transportation system, normally has complex domain-knowledge constraints. Due to the complex constraints, it can be challenging to design and launch adversarial perturbation practically. For example, the sensing data manipulated with adversarial perturbation may violate the constraints in CPSs and can be detected via built-in detectors that are conventionally designed based on domain-knowledge constraints. To address this challenge, we propose a practical and stealthy adversarial attack where adversarial perturbations are able to sufficiently mislead learning model in CPSs while bypassing the built-in detectors effectively. Concretely, our proposed CPS-specific adversarial attack method delivers three main contributions:

- We propose an unsupervised disentangled representation model to learn and interpret the features of CPSs' sensing data by disentangling the features into domain features, which are related to domain-knowledge constraints, and attribute features that are not highly correlated to the constraints. Using these explainable feature maps, our proposed method can produce practical and stealthy adversarial perturbations.

- Our method provides a novel and practical solution to effectively select and utilize explainable features for synthesizing adversarial perturbations in CPS domain.

- Our proposed method does not require any handcrafted domain knowledge to be integrated explicitly in the attack model formulation. By doing so, the attacker is not required to have sufficient knowledge of the targeted CPS system. Additionally, this also results in a more general application scenario for our method, specially when the domain-knowledge constraints cannot be represented in a mathematically differentiable form.

The rest of the paper is organized as follows. In Section 2, we will review related work. In Section 3, we will introduce our proposed CPS-specific adversarial attack. In Sections 4

and 5, the performance evaluations and conclusions will be presented, respectively.

## 2 Related Work

In this section, we review the state-of-the-art works related to our proposed method.

### 2.1 White-Box and Black-Box Adversarial Attacks

Many previous works focus on generating adversarial perturbations based on the full knowledge of the targeted learning model, called white-box adversarial attack. Fast Gradient Sign Method (FGSM) defines the perturbation following the direction of gradient to maximize the loss function of learning model (Goodfellow, Shlens, and Szegedy 2014). Kurakin et al. (2016) introduced an iterative method to search optimal values for adversarial perturbations. Projected Gradient Descent (PGD) method is another iterative method that implements projected gradient to restrict the scale of searched perturbation (Madry et al. 2017). Some following research work emphasized on how to enhance the computation efficiency of PGD (Shafahi et al. 2019; Zhang et al. 2019; Zhu et al. 2020). Additionally, Moosavi-Dezfooli, Fawzi, and Frossard (2016) proposed a method, called DeepFool, to quantify the robustness of state-of-the-art classifiers and compute adversarial perturbations to compromise the classifiers. Papernot et al. (2016) used the first derivative of feed-forward neural network to compute the adversarial samples. Carlini and Wagner (2017) formulated a new optimization instance with Lagrangian relaxation to bound the perturbation for adversarial training. Dong et al. (2018) proposed momentum-based adversarial attack which has a more stable update gradient direction than previous methods. In many practical scenarios, the attacker does not have access to the target learning model. In this situation, a black-box adversarial attack is necessary. Transfer-based method was proposed to generate adversarial perturbations of a surrogate model to compromise the target learning model in a black-box scenario (Liu et al. 2016; Papernot et al. 2017; Lu, Issaranon, and Forsyth 2017; Dong et al. 2018). Published experiment results show that the adversarial perturbations generated against surrogate models can be effective on compromising the target learning models. In order words, transferability can be maintained.

### 2.2 Adversarial Attack In CPSs

As far as we know, very limited research work has been done to generate practical adversarial perturbations in CPSs. In (Li et al. 2021), a search of optimal adversarial perturbation was considered as an optimization problem where domain-knowledge constraints were carefully and explicitly represented as linear equations or inequalities. The search is an iterative process to find proper perturbation which can both mislead learning model and fulfill the domain-knowledge constraints. This method is effective in the considered scenarios. However, this method requires examination of the integrity of domain-knowledge constraints in

each iteration. Additionally, when constraints can not be represented in linear forms, the effectiveness of this method can be compromised.

### 2.3 Disentangled Representation

Disentangled representation learning focuses on extracting domain-invariant features from example pairs. Different methods have been developed to successfully extract the domain-invariant content features via an autoencoder model structure in CV applications (Lee et al. 2018; Cheung et al. 2014; Mathieu et al. 2016). Adversarial training loss from Generative Adversarial Network (GAN) (Goodfellow et al. 2014) is implemented in the disentangled representation model to enforce the learning of disentangled representations. The work of Lee et al. (2018) considered that content and attribute features are designated as the distinct feature spaces for learning the disentangled representations of images. Additionally, this model assumes that the attribute feature aligns with a prior Gaussian distribution.

## 3 Proposed CPS-Specific Adversarial Attack

In this section, we describe our proposed practical and stealthy CPS-specific adversarial attack.

### 3.1 Problem Formulation

In our work, we specify a threat model of our proposed CPS-specific adversarial attack as follows:

1. The attacker is assumed to have access to partial data of the targeted data-driven CPSs via eavesdropping and querying for information distillation.

2. Considering that the learning model $F$ and the built-in detector $B$ are critical for a data-driven CPS, $F$ and $B$ are always launched with advanced security measures. Therefore, it is reasonable to assume that the attacker does not have access to $F$ and $B$. The attacker can realize information distillation about $F$ by reconstructing a surrogate model $F_s$ with the data obtained via eavesdropping and querying.

3. It is also reasonable to believe that the training procedure of the learning model $F$ is conducted with advanced security measures. Because of it, we assume that the attacker cannot access the training procedure and the data used for the training procedure.

4. The objective of the attacker is to launch an adversarial attack that is able to sufficiently mislead the learning model $F$ during the inference procedure while bypassing the built-in detector $B$ effectively.

5. The attacker has limited knowledge of the physical and mechanical constraints, called domain-knowledge constraints in our paper, of the targeted CPSs.

For simplicity, in our work, the learning model $F$ in the targeted CPS is considered to make classification-based decision making, such as detecting false data injection attack in a power system and detecting vehicle state attack in a transportation system.

To realize a practical and stealthy attack, the adversarial perturbation $r$ should be optimized to maximize the cross-entropy for misleading the target learning model $F$, which is described as:

$$r = \underset{||r||_\infty \leq \varepsilon}{\arg\min} \log_2[P(y|x + r, F)]. \qquad (1)$$

where $x$ and $y$ are a sensing data sample and its associated label, respectively, and $\varepsilon$ is the upper bound of the infinite norm of perturbation $r$. However, since $F$ is unknown to the attacker, the attacker needs to reconstruct a surrogate model $F_s$ for information distillation about $F$. Therefore, Eq. (1) is reformulated as follows:

$$r = \underset{||r||_\infty \leq \varepsilon}{\arg\min} \log_2[P(y|x + r, F_s)]. \qquad (2)$$

In our work, we aim to build an effective surrogate model $F_s$, based on which the adversarial perturbation $r$ is generated to not only mislead the learning model $F$ but also bypass the built-in detector $B$.

## 3.2 Framework Overview

The overview of our proposed framework for generating the adversarial perturbation $r$ is illustrated in Fig. 1. As shown in Fig. 1, the framework mainly consists of three steps:

Step 1: A disentangled model is trained to learn and interpret the features of the sensing data samples by disentangling the features into domain features and attribute features. A completed disentangled model includes two encoders and a decoder. We only utilize the selected encoder in the following steps for generating adversarial perturbations.

Step 2: The selected encoder of the disentangled model is reused as a transfer-learning model to optimize the cascaded discriminator. The encoder and discriminator constitute the surrogate model $F_s$ that is used for information distillation for the learning model $F$.

Step 3: The surrogate model $F_s$ is utilized to generate the adversarial perturbations that have transferability to compromise the target learning model $F$. The attacker queries the target learning model $F$ with the generated adversarial data sample $x + r$ for testing the effectiveness of the attack strategy.

In the next subsections, we detail the main steps of our proposed framework.

## 3.3 Design of Disentangled Representation Model

As stated above, in our proposed framework, a disentangled representation model is designed for interpreting the features of the sensing data by disentangling the features into domain and attribute features, which enables stealthy adversarial attack. Inspired by the work of Lee et al. (2018), we propose a novel autoencoder structure by exploiting cycle-generative adversarial network (CycleGAN) (Zhu et al. 2017) for realizing the disentangled representation model. A data pair $(x_1, x_2)$ sampled from the accessible dataset is fed to the proposed disentangled representation model shown in Fig. 2. Two encoders $E_d$ and $E_a$ are utilized to extract domain and

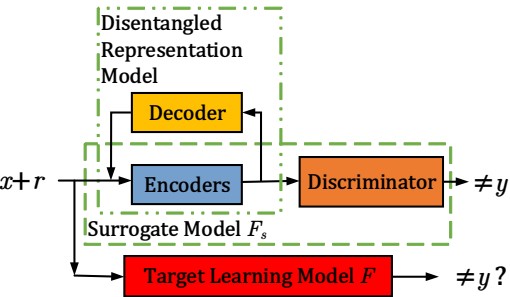

Figure 1: Overview of our proposed framework for generating CPS-specific adversarial attack.

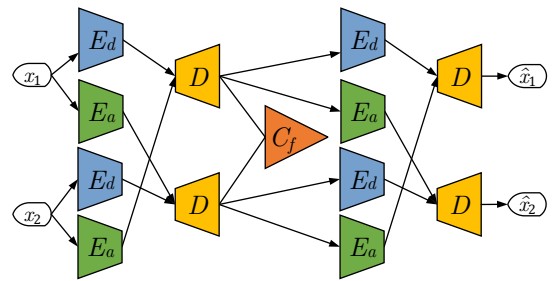

Figure 2: The overview of the proposed disentangled representation model.

attribute features, respectively. After the encoders, the extracted features of the two input data samples are mixed and fed to decoder $D$. The decoder generates two new fake data samples $f_1$ and $f_2$, which are associated with the real data $x_1$ and $x_2$, respectively. A discriminator $C_f$ is implemented after the decoder to identify whether the input data is real or fake. The generated fake data are then encoded and decoded again in the same mixing pattern as the previous autoencoder. Therefore, the generated fake examples in the second round should be the same as or very close to the input data samples if the encoders and decoder are optimal.

During the training process of our proposed disentangled representation model, there are eight loss-function terms required to be minimized as stated in the followings:

1 Cycle-reconstruction loss: this loss represents the mean square error between the generated fake data $\hat{x}$ in the second round and the associated real data $x$:

$$L_1 = ||\hat{x}_1 - x_1||_2^2 + ||\hat{x}_2 - x_2||_2^2,$$

$$\text{where} \begin{cases} \hat{x}_1 = D(E_d(f_1), E_a(f_2)) \\ \hat{x}_2 = D(E_d(f_2), E_a(f_1)) \\ f_1 = D(E_d(x_1), E_a(x_2)) \\ f_2 = D(E_d(x_2), E_a(x_1)) \end{cases}. \qquad (3)$$

2 Discriminator loss: this loss is based on the binary cross-entropy method. If the data sample is real, the output is one. Otherwise, the output is zero. The loss can be formulated as:

$$\begin{aligned} L_2 = & -\log_2(C_f(x_1)) - \log_2(C_f(x_2)) \\ & - \log_2(1 - C_f(f_1)) \\ & - \log_2(1 - C_f(f_2)). \end{aligned} \qquad (4)$$

3. Adversarial training loss: this loss represents the quality of the fake data samples generated by the autoencoder. The autoencoder belongs to generative model in GAN that performs adversarial optimization. In this case, the autoencoder aims to produce higher-quality fake data samples that are able to bypass the discriminator. The loss is formulated as:

$$L_3 = -\log_2(C_f(f_1)) - \log_2(C_f(f_2)). \quad (5)$$

4. Conditional-reconstruction loss I: if $x_1$ and $x_2$ have different domain features and share the same attribute features, the fake example $f_1$ should be the same as or close to $x_1$. The same situation applies to $f_2$ and $x_2$. Therefore, the loss is used to optimize the model only if $x_1$ and $x_2$ share the same attribute features. The loss can be calculated as:

$$L_4 = ||f_1 - x_1||_2^2 + ||f_2 - x_2||_2^2. \quad (6)$$

5. Conditional-reconstruction loss II: if $x_1$ and $x_2$ share the same domain features and have diverse attribute features, the fake data sample $f_2$ should be the same as or close to $x_1$. The same situation applies to $f_1$ and $x_2$. Therefore, the loss is used to optimize the model only if $x_1$ and $x_2$ share the same domain features. The loss can be calculated as:

$$L_5 = ||f_2 - x_1||_2^2 + ||f_1 - x_2||_2^2. \quad (7)$$

6. Cycle consistency loss: this loss represents the summation of the mean square error between the encoded features in the first round and the encoded features in the second round:

$$L_6 = ||E_d(f_1) - E_d(x_1)||_2^2 + ||E_d(f_2) - E_d(x_2)||_2^2$$
$$+ ||E_a(f_1) - E_a(x_2)||_2^2 + ||E_a(f_2) - E_a(x_1)||_2^2. \quad (8)$$

7. Conditional pair loss I: if $x_1$ and $x_2$ share the same domain features and have diverse attribute features, $E_d(x_1)$ and $E_d(x_2)$ should be the same or close to each other. Additionally, $E_a(x_1)$ and $E_a(x_2)$ should be very different from each other. The loss can be represented as:

$$L_7 = ||E_d(x_1) - E_d(x_2)||_2^2 - ||E_a(x_1) - E_a(x_2)||_2^2. \quad (9)$$

8. Conditional pair loss II: if $x_1$ and $x_2$ share the same attribute features and have diverse domain features, $E_a(x_1)$ and $E_a(x_2)$ should be the same or close to each other and the domain features $E_d(x_1)$ and $E_d(x_2)$ should be very different from each other. Thus, the loss can be represented as:

$$L_8 = ||E_a(x_1) - E_a(x_2)||_2^2 - ||E_d(x_1) - E_d(x_2)||_2^2. \quad (10)$$

### 3.4 Surrogate Model Construction

Once the disentangled representation model is well-trained, we utilize the encoders $E_d$ and $E_a$ as the initial model to transfer domain knowledge to the surrogate model. The outputs of $E_d$ and $E_a$ are concatenated and fed to a new discriminator $C_c$ to learn the task of the target learning model. Figure 3 illustrates the structure of the surrogate model. Since domain and attribute encoders are well-trained in the previous step, their parameters are fixed and only the parameters of the following discriminator $C_c$ are updated during the training of surrogate model. In other words, the construction of surrogate model can be viewed as a transfer-learning process where a classifier is built on the existing model. Since the attacker is assumed to have no knowledge

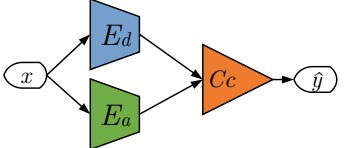

Figure 3: Network structure for surrogate model.

of the target learning model $F$ as stated in our threat model, the discriminator $C_c$ can have an arbitrary structure.

### 3.5 Generation of Adversarial Perturbation

In our proposed method, the generation of adversarial attack based on the surrogate model is realized by exploiting gradient-based algorithms. Three gradient-based algorithms are considered in our current work, including Fast Gradient Sign Method (FGSM) (Goodfellow, Shlens, and Szegedy 2014), a Momentum Iterative Fast Gradient Sign Method (MI-FGSM) (Dong et al. 2018), and Projected Gradient Descent (PGD) (Madry et al. 2017). The loss of generating adversarial perturbation is formulated to maximize the difference between the model prediction $\hat{y}$ and the associated true label $y$, which is shown as follows:

$$L_a = -distance(\hat{y}, y), \quad (11)$$

where $\hat{y} = F_s(x)$. Additionally, to enable a stealthy adversarial attack, the adversarial perturbation needs to be able to bypass the built-in detector that is conventionally designed based on domain-knowledge constraints. Considering that domain features characterize the inherent features associated with domain-knowledge constraints, the impact of the adversarial perturbation on the domain features needs to be minimized. To realize this goal, in our proposed method, the domain encoder $E_d$ is selected to be detached when executing backpropagation on the model $F_s$ for generating adversarial perturbation. By doing so, the gradient calculation is restricted to only follow the direction of extracted attribute features.

## 4 Performance Evaluations

In this section, we will evaluate the performance of our proposed CPS-specific adversarial attack from the perspectives of efficiency on misleading the target learning model and the stealthiness via bypassing the built-in detector effectively. We consider two CPS case studies for performance evaluations: one is about detecting false data injection attack in a

power system, and the other is about detecting vehicle state attack in a transportation system.

## 4.1 Case Study I

In this case study, we consider a CPS scenario where a learning model $F$ is deployed to detect false data injection attack (Liu, Ning, and Reiter 2011) in IEEE 39-Bus System that has 10 generators and 46 power lines (Athay, Podmore, and Virmani 1979). Additionally, a built-in residual-based detector is considered, which is formulated as follows:

$$||z + a - Hs'||_2 \leq \alpha. \quad (12)$$

where $a$ denotes an attack vector, $z$ denotes the original benign measurement data, $H$ denotes the a constant matrix characterizing physical constraints of the power system, $s'$ is the state estimation based on the measurements potentially compromised by the attack vector $a$, and $\alpha$ is the threshold of the built-in detector. Using our proposed method, an adversarial attack is generated to mislead the learning model $F$ sufficiently while bypassing the built-in detector formulated in Eq. (12) effectively.

In our case study, the sensing dataset are collected from simulations with multiple load profiles and two topology profiles. We consider the constraints represented by the topology profiles correspond to the domain features in our disentangled representation model. Additionally, the information represented by load profiles corresponds to the attribute features. Each training data sample pair for training the disentangled representation model is randomly sampled from the dataset. If the pair shares the same load profile, conditional-reconstruction loss I and conditional pair loss II are applied. If the pair shares the same topology profile, conditional-reconstruction loss II and conditional pair loss I are applied. All the other four loss-function terms defined in Section 3.3 are always applied no matter which pair is fetched for optimizing the model. In addition, 2000 data samples are collected, of which 900 data samples are used for realizing the learning model $F$ for detecting the false data injection attack in the power system and 1100 data samples are considered to be accessible to the attacker for developing and deploying adversarial attack. Furthermore, in our case study, we consider the target learning model $F$ is in the form of either fully connected neural network (FCNN) or recurrent neural network (RNN).

For generating the adversarial perturbation, we first train a RNN-based disentangled model to extract the domain and attribute features. Then the encoders of this model is utilized as a pre-trained model to fine-tune a surrogate model. The training data samples for the surrogate model include the benign data samples and the data manipulated via false data injection attack. The domain encoder is selectively detached during the calculation of the gradient for perturbation generation. As a comparison, we also introduce general gradient-based adversarial attack methods as baseline methods. The success rates of our proposed CPS-specific adversarial attack on misleading the decision making by exploiting FGSM, MI-FGSM and PGD method are shown in Figs. 4 to 6, respectively. As illustrated in the plots, our proposed method is able to achieve comparable success rates compared with the baseline methods. Additionally, our proposed method shows better transferability on misleading FCNN-based target learning model compared with misleading RNN-based learning model. We continue to evaluate the capacity of our proposed adversarial attack on bypassing the built-in detector. The bypassing capacity $g$ is formulated as:

$$g = \min\Big(\frac{1}{\log_2(||z + a - Hs'||_2)}, 1\Big). \quad (13)$$

The performance evaluation is shown in Fig. 7. From Fig. 7, it is clear that our method outperforms the baseline methods. Additionally from Figs. 4 to 7, we can get that our proposed adversarial attack is able to achieve a good trade-off between the misleading efficiency and bypassing capability.

## 4.2 Case Study II

In this case study, we leverage the Veremi dataset for performance evaluation by considering a scenario of detecting vehicle state attack in a transportation system (van der Heijden, Lukaseder, and Kargl 2018). The dataset collects the vehicle state message from 37500 vehicles in Luxembourg SUMO Traffic scenario (Codecá et al. 2017). The vehicle state message contains the positions of transmitter and receiver, speed, and elapsed time. The injected vehicle state attack includes constant value, constant offset, random value, random offset, and eventual stop (van der Heijden, Lukaseder, and Kargl 2018). In this transportation system, a deep learning model is deployed to detect the vehicle state attack. Additionally, the system also has two types of built-in detectors: sudden appearance warning (SAW) based detector and acceptance range threshold (ART) based detector. SAW-based detector detects the vehicle state attack by measuring the moved distance between samples and identifying the unreasonable moving distance. ART-based detector detects the attack by estimating the communication distance between the transmitter and receiver and identifying the unreachable vehicles.

In this scenario, our proposed method firstly trains a RNN-based disentangled model to extract the domain and attribute features. Since no prior domain knowledge is assumed in this scenario, we consider the common features shared by all the data as the domain features and the other features as the attribute features. In this case, all the loss-function terms except conditional-reconstruction loss I and conditional pair loss II are implemented in this scenario. Secondly, we implement a RNN-based surrogate model. We evaluate the performance of our method by considering that the target learning model is based on either FCNN or RNN, which is shown in Fig. 8. From the plots, it is clear that our method outperforms the baseline method, general FGSM, in misleading both FCNN-based and RNN-based target learning models. We also evaluate the capability of our method on bypassing the SAW-based and ART-based detectors. The bypassing capacity of the adversarial attack is formulated as the probability of not being detected by the built-in detectors. The performance evaluation is shown in Fig. 9, from which we can observe that our method outperforms the baseline method especially for ART-based detector. Furthermore, in this case study, we also include a method of detaching

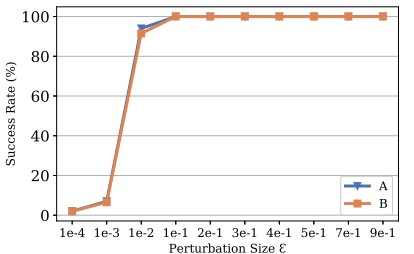 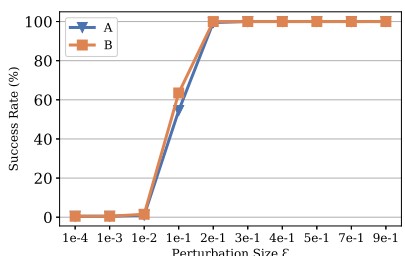 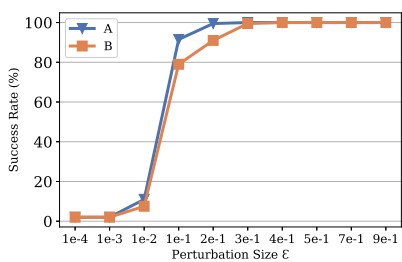

(a) Effectiveness on misleading the surrogate model.

(b) Effectiveness on misleading the FCNN-based target learning model.

(c) Effectiveness on misleading the RNN-based target learning model.

Figure 4: Success rates of adversarial attack based on FGSM: A: General FGSM and B: Our method.

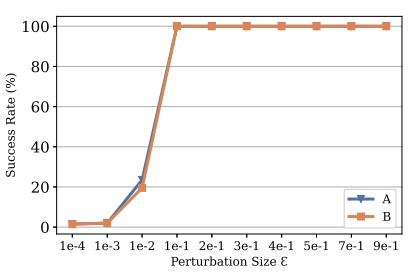 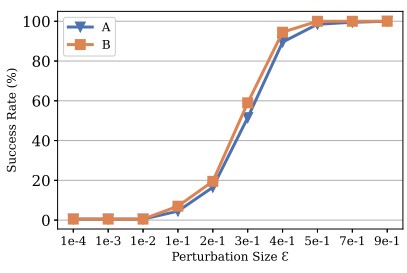 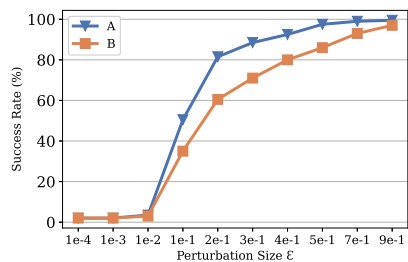

(a) Effectiveness on misleading the surrogate model.

(b) Effectiveness on misleading the FCNN-based target learning model.

(c) Effectiveness on misleading the RNN-based target learning model.

Figure 5: Success rates of adversarial attack based on MI-FGSM where the decay factor $\mu$=1: A: General MI-FGSM and B: Our method.

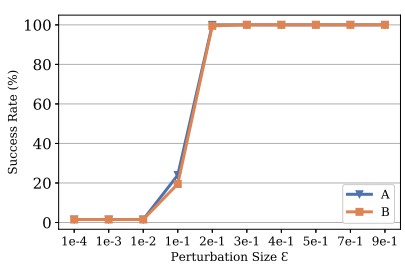 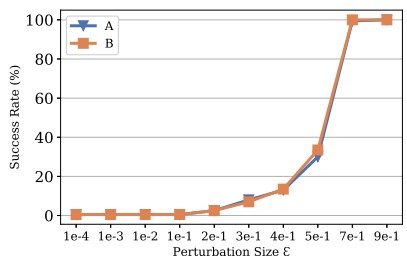 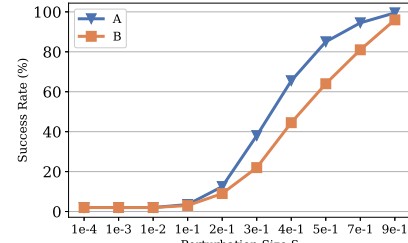

(a) Effectiveness on misleading the surrogate model.

(b) Effectiveness on misleading the FCNN-based target learning model.

(c) Effectiveness on misleading the RNN-based target learning model.

Figure 6: Success rates of adversarial attack based on PGD: A: General PGD and B: Our method.

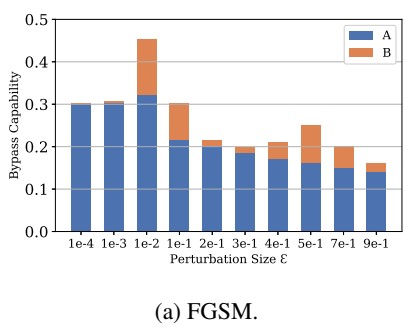 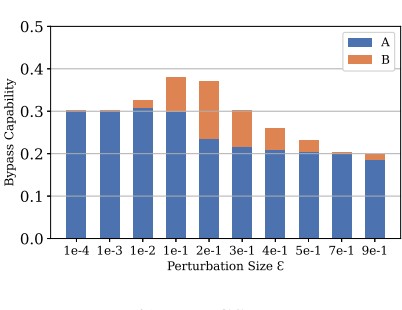 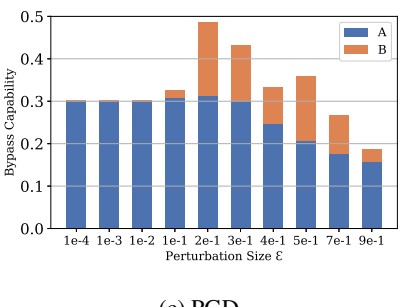

(a) FGSM.

(b) MI-FGSM.

(c) PGD.

Figure 7: Bypass capacities of adversarial attack: A: Baseline method and B: Our method.

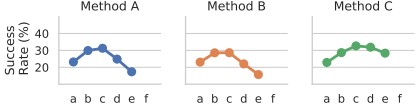

(a) Effectiveness on misleading the surrogate model with different perturbation sizes $\varepsilon$.

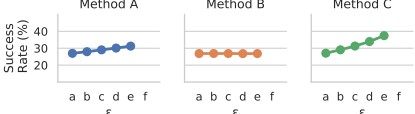

(b) Effectiveness on misleading the FCNN-based target learning model with different perturbation sizes $\varepsilon$.

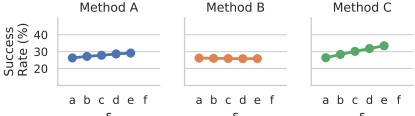

(c) Effectiveness on misleading the RNN-based learning model with different perturbation sizes $\varepsilon$.

Figure 8: Success rates of adversarial attack: A: Baseline method, B: Attribute-encoder detached method, and C: Our method; perturbation size, $\varepsilon$ - a: 0, b: 0.0001, c: 0.001, d: 0.002 e: 0.003, f: 0.004.

attribute encoder for evaluation purpose. Since our method detaches the domain encoder during calculating adversarial perturbation, attribute-encoder detached method is considered as a comparison to illustrate the importance of selectively detaching domain encoder in our design. The performance comparisons between our method and the attribute-encoder detached method shown in Figs. 8 and 9 illustrate the the effectiveness of our proposed method to interpret the data features by extracting domain and attribute features and selectively utilize explainable features for generating adversarial perturbations.

## 5 Conclusions

In this paper, we propose a novel CPS-specific adversarial attack method that is able to compromise the learning model of a data-driven CPS in a practical and stealthy manner. Our work presents three main contributions. Firstly, our method enables an unsupervised disentangled representation model for learning and interpreting the data features by disentangling the features into domain features and attribute features. Using the obtained explainable feature maps, it is feasible to produce practical and stealthy adversarial perturbations. Secondly, our work provides a novel approach to synthesize adversarial perturbations where explainable features are selectively utilized, which leads to a more practical adversarial attack. Thirdly, our adversarial attack method does not require any explicit integration of domain-knowledge constraints in attack model formulation, resulting in more general application scenarios especially when the attacker

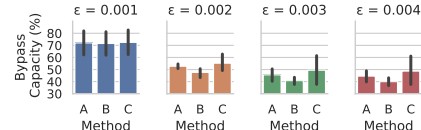

(a) Bypass capacities considering SAW-based detector at 50 meters with different perturbation sizes $\varepsilon$.

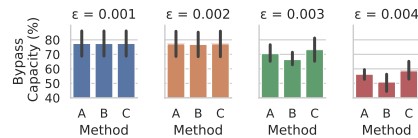

(b) Bypass capacities considering SAW-based detector at 100 meters with different perturbation sizes $\varepsilon$.

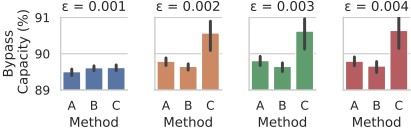

(c) Bypass capacities considering ART-based detector at 200 meters with different perturbation sizes $\varepsilon$.

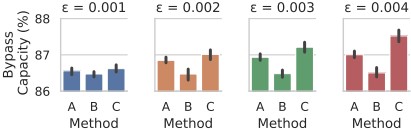

(d) Bypass capacities considering ART-based detector at 300 meters with different perturbation sizes $\varepsilon$.

Figure 9: Bypass capacities of adversarial attack. Method - A: Baseline method, B: Attribute-encoder detached method, and C: Our method.

has limited knowledge of the targeted CPSs or the domain-knowledge constraints cannot be represented in a mathematically differentiable form. As illustrated in the simulation results, our proposed method is able to sufficiently mislead the learning model in the target CPSs while effectively bypassing the built-in detector that is normally designed based on physical and mechanical constraints of the CPSs. In our ongoing work, we are working on evaluating our proposed method in other CPS domains and exploring a more general form of our proposed adversarial attack which can be suitable for various CPS applications.

## 6 Acknowledgements

This work was supported by an Early Career Faculty grant from NASA's Space Technology Research Grants Program.

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
