# OpenReview forum: "A Practical and Stealthy Adversarial Attack for Cyber-Physical Applications"
_AAAI.org/2022/Workshop/AdvML — AAAI-22 AdvML Workshop LongPaper_

### Official Review · Reviewer_gS8q · 2021-11-27
**A good paper to realize adversarial attacks in CPS scenario. It has been shown effective in two defense cases, compared with baseline models.**

**Rating:** 7
**Confidence:** 4

**Review:**

Pros:
1. This paper utilizes Cycle-GAN to disentagle domain and attribute features. This may be something new in CPSs.
2. The paper is easy to follow.
3. The experiment results are sufficient with two cases involving safety-sensitive applications.
Cons
1. The paper is in lack of novelty. It is more like a application of latent space attack in CPS scenario.

---

### Official Review · Reviewer_4go9 · 2021-11-30
**Adversarial attacks for cyber-physical system.**

**Rating:** 7
**Confidence:** 4

**Review:**

This paper proposes a novel adversarial attack method for cyber-physical systems (CSPs). An encoder-decoder model is first trained to learn and interpret the features of sensing data. The encoder is further used to train a discriminator. Finally, the encoder and discriminator constitute a surrogate model to generate adversarial examples. Two CSP case studies are considered to demonstrate the effectiveness of the proposed method.

Weakness:

It’s better to introduce the CPS in related work.

More advanced transfer-based attack methods should be considered since FGSM and PGD are not strong transfer-based attacks, such as momentum-based attack methods [1].

[1] Dong, Y., Liao, F., Pang, T., Su, H., Zhu, J., Hu, X., & Li, J. (2018). Boosting adversarial attacks with momentum. In Proceedings of the IEEE conference on computer vision and pattern recognition (pp. 9185-9193).

---

### Decision · Program_Chairs · 2021-12-01

**Decision:**

Accept (Long Paper)

**Comment:**

Both reviewers agree to accept this paper. Please address the reviewers' comments in camera-ready.